# A Comprehensive Review of Electromagnetic Interference Shielding Composite Materials

**DOI:** 10.3390/mi15020187

**Published:** 2024-01-26

**Authors:** Silvia Zecchi, Giovanni Cristoforo, Mattia Bartoli, Alberto Tagliaferro, Daniele Torsello, Carlo Rosso, Marco Boccaccio, Francesco Acerra

**Affiliations:** 1Department of Applied Science and Technology, Politecnico di Torino, Corso Duca degli Abruzzi 24, 10129 Torino, Italy; silvia.zecchi@polito.it (S.Z.); giovanni.cristoforo@polito.it (G.C.); daniele.torsello@polito.it (D.T.); 2Consorzio Interuniversitario Nazionale per la Scienza e Tecnologia dei Materiali (INSTM), Via G. Giusti 9, 50121 Firenze, Italy; mattia.bartoli@polito.it; 3Italian Institute of Technology, Via Livorno 60, 10144 Torino, Italy; 4Istituto Nazionale di Fisica Nucleare, Sez. Torino, Via P. Giuria 1, 10125 Torino, Italy; 5Department of Mechanical and Aerospace Engineering, Politecnico di Torino, Corso Duca degli Abruzzi 24, 10129 Torino, Italy; carlo.rosso@polito.it; 6Leonardo Labs, OGR Tech, Corso Castelfidardo 22, 10138 Torino, Italy; 7Leonardo Aircraft, Viale dell’Aeronautica Sns, 80038 Pomigliano d’Arco, Italy; francesco.acerra@leonardo.com

**Keywords:** composite materials, microwave shielding, microwave absorption

## Abstract

The interaction between matter and microwaves assumes critical significance due to the ubiquity of wireless communication technology. The selective shielding of microwaves represents the only way to achieve the control on crucial technological sectors. The implementation of microwave shielding ensures the proper functioning of electronic devices. By preventing electromagnetic pollution, shielding safeguards the integrity and optimal performances of devices, contributing to the reliability and efficiency of technological systems in various sectors and allowing the further step forwards in a safe and secure society. Nevertheless, the microwave shielding research is vast and can be quite hard to approach due to the large number and variety of studies regarding both theory and experiments. In this review, we focused our attention on the comprehensive discussion of the current state of the art of materials used for the production of electromagnetic interference shielding composites, with the aim of providing a solid reference point to explore this research field.

## 1. Introduction

Electromagnetic interference (EMI) shielding materials are crucial in modern electronic devices, as they prevent electromagnetic waves from interfering with the proper functioning of the devices [1,2]. These materials are usually composed of conductive or magnetic particles, which form a barrier that absorbs or reflects the electromagnetic waves. In recent years, the research in this field has been focused on finding new and improved materials for EMI shielding, which can provide better performance, a lower cost, and better compatibility with several electronic devices [3,4,5,6,7].

Composite materials have emerged as a prominent category of materials that have garnered considerable attention in the realm of EMI shielding [8]. By combining different materials, composites can exhibit superior characteristics that surpass those of individual components alone [9]. The choice and arrangement of the constituents in composite materials play a crucial role in determining their EMI shielding effectiveness (EMI SE) [10]. Factors such as the type, size, and concentration of fillers, as well as the matrix material that binds them, can significantly influence the overall shielding performance [11]. The distribution and orientation of the fillers within the matrix also impact the ability of the composite to block or absorb electromagnetic waves [12,13].

Besides their shielding effectiveness, other important characteristics to consider when selecting composite materials for EMI shielding include mechanical properties, thermal stability, and durability [14]. The composite should possess sufficient mechanical strength to withstand the stresses and strains encountered during fabrication and device operation [11]. It should also exhibit good thermal stability to ensure its performance is not compromised under varying temperature conditions [15]. Additionally, the composite should be durable, resistant to environmental factors, and capable of maintaining its shielding properties over time [15].

Several techniques are employed to fabricate EMI shielding composites. Common methods include solution blending, melt blending, and in situ polymerization. Solution blending involves dispersing the fillers in a solvent and mixing them with the matrix material, followed by the removal of the solvent [16]. Melt blending involves the direct mixing of fillers and the molten matrix material, which is then solidified to form the composite [17]. In situ polymerization involves the formation of the composite material during the polymerization process itself [18]. Nonetheless, a significant drawback of this method is the tendency of nano-fillers to aggregate, which poses a considerable challenge [19].

Advancements in nanotechnology have further contributed to the development of innovative composite materials for EMI shielding [20]. Nanocomposites, which incorporate nanoscale fillers such as nanoparticles or nanofibers, offer enhanced properties due to their high surface-area-to-volume ratio and quantum size effects [21]. These nanofillers can improve the conductivity, permeability, and mechanical strength of the composite, leading to improved EMI shielding performance.

The fillers used for the preparation of EMI shielding composites can be classified into three categories: organic, inorganic, and hybrid fillers [22]. Organic fillers (i.e., carbon black, graphene, carbon nanotubes) are lightweight and highly conductive, enabling their use for flexible electronics [23]. Carbon black (CB) is a conductive filler that can provide EMI shielding in flexible electronics. Graphene, a two-dimensional material, has excellent electrical conductivity and can provide high-performance EMI shielding. Carbon nanotubes (CNTs), which are composed of rolled-up sheets of graphene, have also shown promise in EMI shielding due to their high conductivity and light weight. However, the high cost and difficulty of synthesizing these materials remain significant challenges. For these reasons, there have been remarkable efforts to create cost-effective carbon-based fillers from organic waste, all while maintaining the standards for high performance. In this sustainable view, biochar is a promising candidate as its properties are effective and tunable [24,25].

Inorganic fillers (e.g., metal powders, metal oxides) provide high conductivity and magnetic permeability, making them suitable for use in high-performance devices [26]. Metal powders, such as silver, copper, and nickel, are widely used in EMI shielding materials due to their excellent conductivity. However, they are expensive and can be challenging to integrate into electronic devices. Metal oxides, such as ferrites and iron oxide, are also commonly used in EMI shielding materials due to their high magnetic permeability. They are relatively inexpensive and easy to integrate into electronic devices, making them an attractive option for shielding materials [5,6].

The integration of inorganic nanoparticles into polymer matrices, as seen in Polymer-Inorganic NanoComposites (PINCs), has emerged as a promising approach for developing advanced EMI shielding materials [27]. By incorporating conductive or magnetic nanoparticles into a polymer matrix, PINCs can effectively attenuate and block electromagnetic waves, providing enhanced EMI shielding capabilities [28]. These materials combine the advantages of inorganic nanoparticles, such as high conductivity or magnetic properties, with the flexibility, lightweight nature, and processability of polymers [1]. This allows for the design and fabrication of EMI shielding materials with improved performance and desirable properties.

Researchers have been exploring various types of conductive or magnetic nanoparticles, including metals, metal alloys, carbon-based materials, and oxides, to achieve optimal EMI shielding effectiveness [20]. Additionally, advancements in nanotechnology have enabled the precise control of nanoparticle dispersion and orientation within the polymer matrix, further enhancing the EMI shielding performance of PINCs [29]. Moreover, the compatibility of PINCs with different fabrication techniques, such as solution-based processes or additive manufacturing methods, offers flexibility in producing EMI shielding materials tailored for specific device requirements.

Hybrid fillers, formally a combination of both organic and inorganic materials, can take advantage of the strengths of both types of fillers while mitigating their respective limitations. The organic fillers contribute flexibility, light weight, and compatibility with flexible electronic devices, while the inorganic fillers provide high conductivity and magnetic permeability. For example, hybrid fillers can be designed to have specific magnetic or electrical properties, depending on the application [30,31].

EMI shielding materials design is a complex field and several requirements must be considered, such as electrical conductivity, magnetic permeability, and cost [30]. The conductivity of the material is critical, as it determines the ability of the material to absorb or reflect electromagnetic waves [32]. Magnetic permeability is also important, as it affects the effectiveness of the shielding material in blocking the electromagnetic waves [33]. Additionally, the cost of the material is a key factor affecting the overall cost of the electronic device. Additionally, EMI shielding materials should not interfere with the proper functioning of the device without significantly affecting the device’s weight or volume [34].

The complexity of EMI shielding materials’ production requires critical insight from several fields, ranging from applied physics to materials science. Accordingly, we comprehensively review the most advanced scientific and technical achievements in the field of EMI shielding composite materials’ production and applications, aiming to provide a reference point for both the experts of the field and the newcomers.

## 2. Microwave Shielding

The interaction between matter and microwaves is related to the components of the matter (atoms), and to the presence of charge carriers (free electrons) and magnetic dipoles [35]. The behavior is due to both the electric and magnetic properties of matter, in particular electrical conductivity, dielectric permittivity and magnetic susceptibility [36]. When microwaves interact with matter, they cause the electrons and magnetic dipoles to oscillate and generate, respectively, an oscillating electric current (and consequentially an electric field) and an oscillating magnetic field within the material, which are in phase with the incident microwave. The magnitude of these fields depends on the conductivity and susceptibility of the material [37]. From a macroscopic point of view, microwave–matter interaction can be expressed using Maxwell’s equations (Equations (1)–(4)). These provide the theoretical foundation for understanding microwave interaction with matter and they are essential for predicting and analyzing the behavior of electromagnetic fields in different materials. Maxwell’s equations for microwave–matter interactions of monochromatic sinusoidal waves are as follows:(1)$$\nabla \cdot \varepsilon E=\rho_{f}$$
(2)∇·μH=0
(3)∇×E+jωμH=0
(4)$$\nabla \times H-j\omega \varepsilon E=J_{f}$$
where E and H are the electric and magnetic fields, *ρ_f_* is the charge density, *ε* is the complex permittivity, µ is the complex permeability, ω is the angular frequency, *J_f_* is the free current, and j is the imaginary unit [36]. The interaction of the material with microwaves is related to its electrical (complex permittivity) and magnetic (complex permeability) properties. When an electromagnetic wave interferes with matter, the following different phenomena occur, as shown in Figure 1: reflection (and secondary reflection), absorption, internal multiple reflection, and transmission.

Reflection arises when a portion of the incident microwave is reflected backward from the material. It occurs when there is a change in the impedance between two materials. Impedance is a fundamental electrical property describing the opposition that a material (or a device) offers to the flow of an alternating current or a time-varying signal. It is a complex quantity typically denoted as “Z”. It can be expressed using Ohm’s law as the ratio of phasor voltage V to phasor current I in an AC circuit: Z = V/I. Impedance plays a crucial role in microwave–matter interaction: when the microwave passes from one material to another with a different impedance (e.g., from air to a generic solid), the electric field of the microwave experiences a change in the direction. This causes some of the wave to be reflected back. The amount of this reflection depends on the magnitude of this impedance mismatch, on the angle of incidence of the wave and on its s or p polarization [38]. Absorption occurs when a wave passes through matter: the interaction between wave and matter constituents causes the transfer of part of its energy to the material, which is usually converted in heat generation. Absorption is a complex phenomenon made of several dissipation mechanisms. They are divided into dielectric losses, magnetic losses and conductive losses; depending on the local composition of the matter that interacts with the wave, one or more mechanisms occur, dispersing the energy of the radiation. Absorption mechanisms can be related to constitutive parameters of the material (dielectric permittivity, magnetic permeability and conductivity) [35,39,40]. Internal multiple reflection is the reflection that occurs between the two walls of the shield. It is favored by the presence of several internal interfaces, such as nanostructures and inhomogeneities within the material [41]. Interactions between wave and inclusion can generally result in scattering phenomena and subsequent diffusion inside the material. The event of a scattering interaction depends on the wavelength of the radiation and on the size of the inclusion. Normally, the contribution of internal multiple reflection is negligible. It is worth noting that certain structures, such as foams or porous materials, exhibit this behavior (multiple reflection). Within these materials, internal structures can become resonant cavities, leading the materials to better shielding properties. It has been reported to be a strong absorption mechanism for these materials [42].

All these contributions cause energy loss of the incident waves and only a reduced portion of the wave passes through the thickness of the material. Electromagnetic shields are aimed to maximize these phenomena, absorbing or reflecting the unwanted signals and reducing their intensity, in order to insulate a portion of space from electromagnetic pollution.

The ability of a material to shield electromagnetic waves can be expressed as electromagnetic interference shielding effectiveness (EMI SE) or shielding effectiveness (*SE_T_*). It evaluates the material shielding properties and it is expressed as the logarithm of the ratio between transmitted (*t*) and incident (*i*) power *P*. *SE_T_* can be also expressed as the ratio between electrical (*E*)/magnetic (*H*) transmitted and incident field (Equation (5)).
(5)$${SE}_{T}\left(dB\right)=10\log_{10}\frac{P_{t}}{P_{i}}=20\log_{10}\frac{E_{t}}{E_{i}}=20\log_{10}\frac{H_{t}}{H_{i}}$$

Total shielding effectiveness can be also expressed as the sum of the shielding contributions (absorption *A*, reflection *R* and internal multiple reflection *M*) as reported in Equation (6):(6)$${SE}_{T}\left(dB\right)={SE}_{A}\left(dB\right)+{SE}_{R}\left(dB\right)+{SE}_{M}\left(dB\right)$$

Generally, the internal multiple reflection contribution is not very relevant and can be neglected, unless a material is engineered to maximize it. The evaluation of these parameters is essential in order to understand the shielding mechanism. Researchers have developed numerous models for the estimation of absorption, reflection and multiple reflection contributes. Here, we report an example of a model for every phenomenon [43]. This model has been widely used in a large number of works for EMI SE evaluation because both dielectric and magnetic components are considered. In this view, it is useful for a comprehensive comparison between different types of composite materials, whether magnetic, dielectric or hybrid. The decomposition of total shielding effectiveness into the individual contributions is shown below. Absorption losses (Equation (7)) can be expressed as follows:(7)$${SE}_{A}\left(dB\right)=8.68t\sqrt{\frac{\omega \sigma_{ac}{µ}_{r}}{2}}$$
where *t* is the thickness of the shield, *ω* is the angular frequency (*ω =* 2*πf* where *f* is the frequency of the wave), *σ_ac_* is the electrical conductivity (evaluated as *σ_ac_ = ωε*_0_*ε*″ where *ε*_0_ is vacuum permittivity and *ε*″ is the imaginary part of the permittivity of the shield), and *µ_r_* is the relative magnetic permeability. Absorption is mainly related to three parameters. Firstly, this mechanism is related to the product between conductivity and permeability. Consequently, magneto-conductive materials exhibit good absorption properties. In addition, the thickness of the shield plays a key role since the amplitude of electromagnetic radiation decreases exponentially as it passes through the shield. The contribution of reflection (Equation (8)) can be estimated as follows:(8)$${SE}_{R}(dB)=10\log\frac{\sigma_{ac}}{16\omega {\varepsilon_{0}µ}_{r}}$$

From this expression, it is clear that reflection is proportional to conductivity and inversely proportional to the relative permeability of the shield. Unlike absorption, reflection depends on the ratio *σ_ac_*/µ*_r_*. This means that the greater the magnetic permeability of the shield, the greater the contribution of absorption with respect to reflection. Moreover, the contribution of reflection decreases as frequency increases. The presence of mobile charge carriers enhances the ability of the shield to reflect electromagnetic waves. Multiple reflection losses (Equation (9)) are directly related to absorption losses and they are evaluated as follows:(9)$${SE}_{M}\left(dB\right)=20\log_{10}(1-e^{-\frac{2t}{\delta }})=20\log_{10}(1-10^{\frac{{SE}_{A}}{10}})$$
where *δ* is known as “skin depth”, i.e., the distance required to attenuate a wave to 37% of its power. It can be expressed as reported in Equation (10):(10)$$\delta =8.68(\frac{t}{{SE}_{A}})$$

Multiple reflection is usually negligible. Despite this, some materials are engineered to maximize this type of shielding in order to couple lightness (due to the low density of porous materials) and shielding effectiveness.

As reported, magnetic permeability is important for both absorption and reflection. The magnetic properties of materials play a crucial role in determining the effectiveness of EMI shielding. Magnetic loss arises from material properties. This is due to many phenomena that occur from the microwave–matter interaction, such as eddy current loss, residual loss, hysteresis loss, and domain wall loss. Eddy currents are circulating currents induced in conductive materials by changing magnetic fields. Eddy current loss occurs when these currents encounter resistance in the material and dissipate energy. The heat generated by eddy current losses can be a significant factor in materials used for magnetic shielding. Hysteresis is the lagging of the magnetic field behind changes in the magnetizing force. When a material undergoes cycles of magnetization and demagnetization, energy is lost due to hysteresis. This results in the conversion of magnetic energy into heat during each cycle. Residual loss occurs because of the inability of magnetic dipoles to return to their original state after being subjected to external magnetic fields. It is associated with the residual magnetization left in the material. Moreover, ferromagnetic materials have a resonance frequency at which the alignment of magnetic moments within the material is most easily achieved. This results in an efficient absorption of magnetic energy [44,45,46,47].

When calculating *SE_T_*, *SE_M_* can be neglected if skin depth *δ* is lower than the thickness of the shield [43]. In other terms, for high values of *SE_A_*, the contribution of the internal multiple reflection is negligible [48].

The shielding properties of a device can also be described by using scattering parameters (*S*_ij_ parameters, where i is the entrance port and j the exit one). Electromagnetic measurements in microwave range can be performed by means of a vector network analyzer (VNA). It can characterize the scattering parameters of samples. By processing electromagnetic measurements with appropriate software, you can extrapolate electrical and magnetic properties like electrical conductivity, complex permeability and permittivity. In a two-port system (e.g., VNA), S-parameters are determined by the interaction between the incident waves and the material; more specifically, they describe reflected and transmitted waves. *S*_11_ and *S*_22_ refer to the reflection coefficient measured on port 1 and 2, respectively, whereas *S*_12_ and *S*_21_ are the transmission coefficient [49]. The measurement of S-parameters is performed using a VNA. For a single material, they can be related to reflection *R*, transmittance *T*, and absorbance *A* through Equations (11)–(13):(11)$$R={\left|S_{11}\right|}^{2}={\left|S_{22}\right|}^{2}$$
(12)$$T={\left|S_{12}\right|}^{2}={\left|S_{21}\right|}^{2}$$
(13)$$A=1-R-T=1-{\left|S_{11}\right|}^{2}-{\left|S_{12}\right|}^{2}$$

The knowledge of S-parameters allows us to evaluate the shielding efficiency of a device. Reflection and absorption contributions (Equations (14) and (15)) can be calculated as follows [36]:(14)$${SE}_{R}=10\log_{10}\left(1-{\left|S_{11}\right|}^{2}\right)$$
(15)$${SE}_{A}=10\log_{10}\left(1-\frac{{\left|S_{11}\right|}^{2}}{{\left|S_{21}\right|}^{2}}\right)$$

As illustrated in Figure 2, the microwave region is broad (from 1 GHz to 170 GHz). It is usually divided into sub-bands, characterized by a restricted frequency range. Research studies on EMI shielding mainly focus on a small portion of the entire region, covering frequencies up to 18 GHz. This is due to the large number of devices that find applications in this range. The low-frequency region is used by systems like wireless communications, low-Earth-orbit satellites and GPS (L-band, 1 ÷ 2 GHz), televisions, mobile phone communications and Bluetooth (S-band, 2 ÷ 4 GHz), radio telecommunications and Wi-Fi systems (C-band, 4 ÷ 8 GHz). Higher frequencies are relevant for air traffic control, satellite and space communications and other applications illustrated in Figure 3 [48]. The protection of electronic products from electromagnetic interference is a primary requirement to ensure the proper functioning and safety of the device [30].

## 3. Materials for EMI Shielding

Historically, metal-based materials are the best choice for EMI shielding purposes. As their main drawbacks, they are subjected to corrosion (high maintenance cost), and they are heavy and not easy to form into an adequate shape. More recently, composite materials have been investigated to overcome these problems [51]. As shown in Table 1, Table 2 and Table 3, several solutions have been developed to face EMI shielding in order to reach a trade-off between performances, cost and applicability. The EMI shielding performance of composite materials depends on several factors, such as the dimensionality of the fillers. It varies from 0 to 3 (where 0 refers to particles with all dimensions in the nanoscale, whereas 3 refers to bulk fillers) and influences the properties of the filler itself. Mono-dimensional fillers (e.g., carbon nanotubes) can facilitate the formation of conductive pathways (percolative pathways) that strongly enhance electrical conductivity, contributing to improved EMI shielding performance [52]. Two-dimensional fillers (e.g., graphene, MX-enes) can form percolative pathways too. Moreover, they are characterized by large surface areas that enhance the probability of interaction with electromagnetic waves [53]. Three-dimensional fillers offer shielding throughout the volume. The combination of fillers with different dimensions can lead to synergistic effects. In this respect, hybrid fillers can leverage the multidimensionality of single fillers, offering an enhanced EMI shielding performance [54,55].

As clearly emerged, a target attenuation value of at least −10 dB is required for in-house application, while higher values are required for specific applications. Accordingly, the choice of the composite formulation represents a key factor for the realization of custom solutions [56].

### 3.1. Organic Fillers

Electromagnetic interference shielding performance generally depends on the electrical and magnetic properties of the material. Carbonaceous fillers have negligible magnetic properties; consequentially, their performance depends on conductivity and dielectric properties [57]. A variety of organic nanostructures have been investigated for EMI shielding applications, like carbon nanotubes (single-walled, SWCNTs, and multi-walled, MWCNTs), graphite, graphene, graphene oxide (GO), reduced graphene oxide (rGO), carbon black, conductive polymers, and others. Other advantages of their use are the tunability of their electrical properties (e.g., through the manipulation of their dimensions, physical and chemical structure, etc.) and their dispersibility in polymer matrices (not for all fillers).

**Table 1 micromachines-15-00187-t001:** Organic nanocomposites and their shielding performance.

Filler	Matrix	wt.% Filler	Frequency (GHz)	Max EMI SE (dB)	Main Mechanism of Shielding	Ref.
SWCNTs	PU	0 ÷ 20	8.2 ÷ 12.4	17	Reflection	[58]
SWCNTs	Epoxy resin	0.01 ÷ 15	8.2 ÷ 12.4	>23	-	[59]
MWCNTs	PP	0.5 ÷ 15	8.2 ÷ 12.4	>60		[10]
MWCNTs	Epoxy resin	1 ÷ 10	2 ÷ 20	-	Reflection	[60]
MWCNTs	Epoxy resin	1 ÷ 10	1 ÷ 26.5	-	Absorption	[61]
MWCNTs	PANI	5 ÷ 25	12.4 ÷ 18	39.2	Reflection	[62]
MWCNTs	Epoxy resin	0.125 ÷ 2	8 ÷ 12	-	Absorption	[63]
MWCNTs	Epoxy resin	0.1 ÷ 0.5	12 ÷ 18	16	-	[64]
CNTs sponge	Epoxy resin	0.42 ÷ 2	8.2 ÷ 12.4	33	Absorption	[65]
MWCNTs	E-glass fabric + epoxy resin	up to 4.15	8.2 ÷ 12.4	18.3	Absorption	[66]
Graphite	UHMWPE	0.43 ÷ 7.05 vol.%	8.2 ÷ 12.4	>45	Absorption	[67]
Graphite	ABS	0 ÷ 20	8.2 ÷ 12.4	60	Absorption	[34]
Graphite	EVA/EOC	5 ÷ 30	2 ÷ 4	67.6	Absorption	[68]
Graphite	Epoxy resin	0 ÷ 40	1 ÷ 18	55	Absorption	[69]
Graphene	Epoxy resin	1 ÷ 17	8.2 ÷ 12.4	-	Reflection	[70]
Graphene	Epoxy resin	0.1 ÷ 15	8.2 ÷ 12.4	21	-	[71]
Graphene	PBAT and PLA	Up to 15	8.2 ÷ 12.4	15	Reflection	[72]
rGO	Epoxy resin	1 ÷ 5	8.2 ÷ 12.4	25	Absorption	[73]
rGO	PU foam	0.5 ÷ 6.5	8.2 ÷ 12.4	17	Absorption	[74]
rGO	PS	0.87 ÷ 3.47	8.2 ÷ 12.4	40	Absorption	[75]
3D-rGO	Wax	1 ÷ 10	2 ÷ 18	-	Absorption	[76]
CB	PP/PS blend	10 vol.%	50 MHz ÷ 1.5 GHz	22	-	[77]
CB	EVA/NBR blends	0 ÷ 50 phr	8.2 ÷ 12.4	>28	-	[78]
CB	PA6	10 ÷ 50	8.2 ÷ 12.4	13.4	Absorption	[79]
CB	PI sponge	10 ÷ 50	0.8 ÷ 12	-	-	[80]
CB	EMA/EOC blend	5 ÷ 30	8 ÷ 12	31.4	Absorption	[81]
CB	Chlorinated PE	5 ÷ 30	8.2 ÷ 12.4	38.4	Absorption	[82]
Biochar	UHMWPE	Up to 80	0.3 ÷ 1.5	35	-	[83]
Biochar	Epoxy resin	25	0.1 ÷ 8		-	[84]
Biochar	PVA	1 ÷ 5	8 ÷ 20	>40	-	[85]
Biochar/graphite	PLA	Up to 46 vol.%	18 ÷ 26.5	>30	-	[86]
PANI	PU	-	S-band, X-band	>10 (S-band) >25 (X-band)	Absorption	[87]
PANI	Epoxy resin	5 ÷ 15	2.4 ÷ 8.8	-	Absorption	[88]
PPy@PANI	Wax	50	2 ÷ 18	-	Absorption	[89]
PPy	Silicone	1 ÷ 5	5.85 ÷ 8.2	-	Reflection	[90]
MWCNTs@PPy	Thermoplastic PU	10 40	8.2 ÷ 12.4	>40	Absorption	[91]

#### 3.1.1. Carbon Nanotubes

CNT structures are very attractive for the development of composites as a result of their electrical, mechanical and physical properties. Due to their high aspect ratio, they can easily form a percolative path (i.e., continuity between conductive elements) inside the material that guarantees high electrical conductivity. This electrical path is usually achieved at very low volume fraction for CNTs [92]. A large number of research works have been dedicated to the EMI shielding properties of CNT composites [93,94,95,96,97,98].

Liu et al. [58] developed a polyurethane (PU)/SWCNTs composite in order to couple good mechanical performances (elasticity and impact strength) and EMI shielding properties in X-band (8.2 ÷ 12.4 GHz) using a simple mixing method (solvent casting). They investigated various concentrations of SWCNTs (0 wt.% ÷ 20 wt.%) and they found a very low percolation threshold (0.2 wt.%). The maximum value of EMI SE was achieved by a composite containing 20 wt.% of filler (≈17 dB). Through the calculation of EMI SE contributions, they demonstrated the reflection-dominated mechanism of shielding.

SWCNTs were further investigated by Huang et al. [59]. The authors mixed epoxy resin with three different types of filler (loading of 0.01 wt.% ÷ 15 wt.%): short-SWCNTs, long-SWCNTs and annealed SWCNTs (obtained through the annealing of short-SWCNT at 1100 °C for 3 h in N_2_ atmosphere). Electromagnetic measurements were recorded in X-band. The results highlighted the strong correlation between SWCNTs properties (aspect ratio and structural integrity) and EMI shielding performance: the best outcome belongs to long-SWCNTs that, as a result of their high aspect ratio, easily create a percolation path. The composite that exhibits the highest EMI SE is 15 wt.% long-SWCNTs-epoxy (EMI SE > 23 dB in all X-band).

Excellent results of EMI SE were obtained by Lecocq et al. [10]. They fabricated polypropylene (PP)/MWCNTs composites with different ratios (loading range from 0.5 wt.% to 15 wt.%) via a simple melt blending method. They used models based on scattering parameters (evaluated via a vector network analyzer) to estimate the EMI SE. The materials exhibited outstanding performances in X-band: the best result is achieved by PP + 15 wt.% MWCNTs that showed an EMI SE greater than 60 dB with a very low thickness (1 mm).

Wang and Zhao [60] investigated epoxy resin/MWCNTs (loading from 1 wt.% to 10 wt.%, outer diameter less than 8 nm) composites’ properties in 2 ÷ 20 GHz and found fairly good absorption ratios (related to reflection and transmission). Overall, the absorption ratio is heavily affected by MWCNTs’ loading in the composite.

Nwigboji et al. [61] studied similar systems based on epoxy resin and MWCNTs (with larger outer diameter, ranging from 20 to 30 nm) from 1 to 26.5 GHz, with loadings ranging from 1 wt.% to 10 wt.%. They found trends similar to those of the work previously described, characterized by higher absorption ratios. This study confirmed the dependency of absorption properties with the MWCNT content.

Saint et al. [62] developed MWCNT nanocomposites based on polyaniline (PANI) matrix, a conductive polymer (Figure 4). They prepared different compositions (5, 10, 20, 25 wt.%) and measured electromagnetic parameters in K_u_-band (12.4 ÷ 18 GHz). EMI SE was evaluated by using scattering parameters. The results demonstrated good properties of shielding (Figure 4a): EMI SE values range from 27.5 dB to 39.2 dB, suggesting their effective use in K_u_-band.

Different features of MWCNTs lead to different shielding properties. Che et al. [63] looked for the impact of different dispersion methods (via ball milling and in solution with surfactant aiding) and MWCNT characteristics (diameters and length-to-diameter aspect ratios) on the absorption properties in X-band of epoxy resin/MWCNT nanocomposites with CNT contents from 0.125 wt.% to 2 wt.%. They used three commercially available MWCNTs to produce epoxy/CNTs composites via two different methods. Good absorption (peak value exceeding 97%) and shielding performances were recorded by composites with a thickness of 2 ÷ 3 mm with very low filler contents (0.25 ÷ 0.5 wt.%), loaded with the CNTs with the highest aspect ratio among those available.

The effect of length on the EMI shielding properties of MWCNTs was further investigated by Singh et al. [64]. They developed low-filled epoxy composites (from 0.1 wt.% to 0.5 wt.%) and characterized them in K_u_-band (12 ÷ 18 GHz). As previously described, the aspect ratio is a crucial parameter to ensure good shielding properties; the electromagnetic measurements highlighted a peak of 16 dB (EMI SE) for long-MWCNTs, whereas short-MWCNTs exhibited a maximum peak of 11.5 dB. Furthermore, the mechanical properties were investigated and again the l/d (length/diameter) ratio has a significant role: the flexural strength was improved by 31.6% for composites made with long-MWCNTs (related to neat epoxy resin), whereas short-MWCNTs registered a lower improvement (+18.9%).

An efficient and lightweight EMI shield was realized and tested by Chen et al. [65] who prepared epoxy composites filled with a particular three-dimensional structure. They synthesized a highly porous sponge (porosity > 99%) made of CNTs via chemical vapor deposition (CVD). Then, they infiltrated it with epoxy resin using a vacuum-assisted method and realized three different compositions (0.42 wt.%, 0.66 wt.%, 2 wt.%). Electromagnetic measurements in X-band provided good results; in particular, the best EMI SE was found to be 33 dB for the composite containing just 0.66 wt.% of CNTs sponge. From a mechanical point of view, the use of a 3D structure inside the epoxy matrix provided better properties (+102% for flexural strength, +64% for tensile strength).

In order to obtain a mechanically resistant electromagnetic shield, Vasconcelos da Silva et al. [66] proposed a novel method to infiltrate glass fiber fabrics with an epoxy resin filled with various ratios of CNTs (up to 4.15 wt.%). They made suspensions of CNTs/epoxy/acetone that were used to impregnate E-glass fiber fabrics. They performed electromagnetic measurements in X-band and they found a maximum EMI SE value of −18.3 dB for a composite with very low thickness (2.2 mm).

#### 3.1.2. Graphite and Graphite Derivatives

Graphite is a naturally abundant and inexpensive material characterized by a layered structure of carbon atoms bonded by Van der Waals forces widely used as filler to develop conductive polymer composites [99,100,101,102,103,104]. As previously said for CNTs, composite electrical properties dramatically increase when a continuous path of filler is obtained.

A strategy to develop polymer composites filled with graphite via a facile and green method has been addressed by Jiang et al. [67]. They proposed simple mechanical mixing followed by hot pressing to make UHMWPE (ultra-high-molecular-weight polyethylene)/graphite composites, aiming to create efficient electromagnetic interference shields (Figure 5). The suggested method allows for the creation of a three-dimensional conductive path at very low filler concentration (0.215 vol.%). Electromagnetic measurements in X-band allowed the authors to measure scattering parameters, which were used to calculate the total shielding efficiency. Graphite-filled UHMWPE demonstrated good shielding properties throughout the frequency range; in particular, the material containing 7.05 vol.% of graphite exhibited an EMI SE greater than 45 dB in all X-band.

A similar production method was proposed for the development of acrylonitrile butadiene styrene (ABS)/graphite composites by Sachdev et al. [34], which mixed and hot-pressed graphite and ABS powder. They performed electromagnetic analysis using a VNA in 8 ÷ 12 GHz and found a remarkable EMI SE value of about 60 dB (all in X-band) for the composite with a loading of 15 wt.%. Furthermore, they performed a single analysis of reflection and absorption contributions that showed the predominance of the absorption shielding mechanism over reflection.

Parida et al. [68] synthesized and tested polymer blends of EVA/EOC (ethylene–vinyl acetate/ethylene–co-octene, weight ratio 4:1) composites filled with exfoliated graphite nanoplatelets. Unlike previous research papers, they performed electromagnetic measurements in a different range of frequencies (S-band, from 2 to 4 GHz). They used various loading levels, from 5 wt.% to 30 wt.%. Obviously, the best performance of EMI SE belongs to the more heavily loaded composite (30 wt.%) that exhibits a peak of 67.6 dB at around 2 GHz with a prevailing action of absorption.

A similar filler was used and mixed with epoxy resin to exploit the preparation of an efficient electromagnetic shield over a wide 1 ÷ 18 GHz. Al-Ghamdi et al. [69] synthesized a graphite-based nanofiller (foliated graphite nanosheets, FGN) via the exfoliation of graphite flakes and subsequent ultrasonic irradiation. Electromagnetic analysis demonstrated good shielding properties over all the frequency range. More specifically, epoxy loaded with 40 wt.% FGN exhibited an EMI SE of 35 ÷ 55 dB from L-band to K_u_-band.

##### Graphene

The single constituent of the graphite structure is known as graphene. It is a monolayered material, composed of sp^2^-hybridized carbon atoms bonded in a honeycomb structure. Thanks to its 2D structure (large surface area that guarantees interfacial interaction with matrix) and its mechanical and electrical performances, it is a very promising candidate as filler for EMI shielding polymer nanocomposites.

Abdelal et al. [70] investigated EMI shielding properties of graphene nanoplatelets (GNPs)/epoxy composites loaded with various concentrations (up to 17 wt.%) over X-band. They executed three different sonications that led to GNPs with three different surface areas (300, 500 and 700 m^2^/g). By measuring electromagnetic properties, they were able to apply models to evaluate the shielding properties of reinforced epoxies. The results showed a strong dependency of dielectric constant and EMI SE on GNPs’ surface area and loading. In fact, the best performance was given by epoxy/GNPs 17 wt.%. It exhibited a transmittance T (evaluated as T = |S_12_|^2^, where S_12_ is a measured scattering parameter) lower than 30% over all the X-bands, with a minimum of around 15%.

A similar work was conducted by Liang et al. [71], who produced epoxy/graphene sheet (from 0.1 wt.% to 15 wt.%) nanocomposites for electromagnetic testing. Their materials exhibited a low percolation threshold (0.52 vol.%) and a maximum shielding effectiveness of 21 dB (15 wt.%), indicating their possible application as microwave shielding lightweight composites.

Kashi et al. [72] evaluated the dielectric and EMI shielding effectiveness (in 8 ÷ 12 GHz) of biodegradable polymer nanocomposites (polybutylene adipate terephthalate/GNPs and polylactic acid/GNPs) filled up to 15 wt.%. They determined the single contributions of reflections and absorption in order to investigate the main shielding mechanism. They found a maximum EMI SE of 15 dB at 8 GHz for polylactic acid/graphene 15 wt.% and demonstrated that shielding is mainly due to reflection.

##### Reduced Graphene Oxide (rGO)

There exist graphene-like structures characterized by the presence of functional groups on their surface. Graphene oxide (GO) is a 2D material that differs from graphene due to an abundant number of oxidized groups. Reduced graphene oxide derives from the reduction (chemical, thermal, etc.) of GO. It has been investigated in various studies as a filler for EMI shielding polymer matrices because it exhibits good electrical properties.

Ahmad et al. [73] synthesized a chemically reduced GO and mixed it with an epoxy matrix (loading from 1 wt.% to 5 wt.%). They performed electromagnetic measurements in X-band and applied models to evaluate the shielding properties of their materials. Furthermore, they conducted deep analysis to determine the involved processes of dielectric relaxation. From their calculations, the best performance belongs to epoxy/rGO 5 wt.%, which exhibits an EMI SE of 25 dB at 12 GHz. The trend of EMI SE over frequency is the same for all materials, characterized by a monotonically increasing behavior. The shielding performances of these composites are mainly due to absorption phenomena.

Jiang et al. [74] designed a process to create a flexible and lightweight electromagnetic shielding composite (for X-band) based on a PU foam filled with rGO. They realized a process based on the supercritical CO_2_ foaming of PU, previously coated with rGO. In order to better evaluate the consequences of foaming process, they also realized simple PU/rGO composites. Foaming causes a transition from reflection-dominated shielding to absorption-dominated shielding. Relatively low fillings guarantee good EMI SE. In particular, PU/rGO 3.17 vol.% foam exhibits an EMI SE higher than 17 dB over the whole X-band.

Yan et al. [75] proposed a method to create mechanically robust microwave shielding for X-band. They realized polystyrene (PS)/rGO composites via a high-pressure solid-phase compression molding. The production process led to the formation of a segregated structure characterized by the presence of rGO on the boundaries among PS pellets (Figure 5).

EMI SE was evaluated by measuring the scattering parameters S_11_ and S_21_. They demonstrated the strong dependency of EMI SE on PS particle size: surprisingly, an increase in PS size composites results in higher EMI SE. For composites with loading of 3.5 vol.% and a PS particle size of 3.3 µm, EMI SE is around 20 dB throughout the X-band. The best performance belongs to the same composite (filled with 3.5 vol.% of rGO) with a PS particle size of 96 µm: it exhibited an EMI SE higher than 40 dB (Figure 6).

A honeycomb-like 3D-rGO was produced by Fang et al. [76] through a multistep method (hydrothermal and subsequent thermal treatment). They prepared wax/3D-rGO (1 ÷ 4 wt.%) composites with different thickness (1 ÷ 5 mm) and focused on the evaluation of microwave absorption from 2 GHz to 18 GHz. As a result, 4 wt.% loaded composites exhibited the best performance, characterized by a large reflection loss (RL, it is the amount of electromagnetic energy that is reflected back when it encounters a material) bandwidth. More specifically, the composite with a thickness of 2 mm shows a bandwidth of about 6 GHz (from 6.5 GHz to 12.5 GHz) characterized by a reflection loss lower than −5 dB.

#### 3.1.3. Carbon Black

CB is a carbonaceous nanomaterial extensively used as a filler in the rubber industry. Unlike other carbon-based fillers (i.e., CNTs, graphene), it is very cheap, but its electrical properties hinder its use for EMI shielding devices; in order to produce an efficient EMI shielding composite filled with CB, one should use a filler loading of about 40 wt.%. This impacts other properties of the material (mechanical, processability, etc.) [77]. Many studies have attempted to develop novel strategies capable of lowering the required loading for good shielding properties.

Al-Saleh et Sundararaj [77] realized a PP/PS blend (in various ratios) filled with 10 vol.% CB. They performed electromagnetic measurements at low frequencies (from 50 MHz to 1.5 GHz), involving a narrow portion of the microwave range (L-band). The results demonstrated the dependency of SE on the PP/PS ratio: a pure PP/10 vol.% CB composite exhibits a higher SE (difference of 10 dB) compared with that of pure PS/10 vol.% CB. At microwave frequencies (1 ÷ 1.5 GHz), the best performance of EMI SE has been recorded using pure PP/10 vol.% CB and it is greater than 22 dB.

Rahaman et al. [78] focused on the impact of two different commercially available CB composites on the EMI shielding properties of polymer matrices (ethylene–vinyl acetate EVA, nitrile butadiene rubber NBR and their blends). Composites with various compositions were prepared (0 ÷ 50 phr). Electrical and electromagnetic analyses were performed in X-band. The results demonstrated the relation between the formation of a percolative network and the shielding properties. Furthermore, they highlighted that, for the same filler loading, the conduction properties are better for CB that exhibits a higher tendency to aggregate. An EMI SE greater than 28 dB was exhibited throughout the X-band by the EVA/NBR blends containing 20 phr of CB.

Porous polymer composites made of polyamide 6 (PA6) and CB were developed and tested by Cai et al. [79]. They investigated the EMI shielding properties in the X-band of composites with different loadings (from 10 wt.% to 50 wt.%). By measuring the scattering parameters, they calculated the EMI SE. They found good levels of EMI SE for the 30 wt.%-loaded composite (13.4 dB). These composites exhibit an absorption-dominated shielding mechanism.

As mentioned above, the high filling of polymers with CB compromises the mechanical performances. Kim et al. [80] investigated the possibility of fabricating a composite sponge polyimide (PI)/CB (10 ÷ 50 wt.%) with superior mechanical performances. Through a four-step method, they were able to fabricate composite porous membranes. Subsequently, by hot-pressing the membranes for 30 min at 250 °C, they realized PI/CB sponges. To better understand the effect of hot-pressing, they also prepared non-hot-pressed samples. Electromagnetic measurements were performed from 800 MHz to 12 GHz. The comparison of pressed and unpressed samples demonstrates the impact on the shielding performances, as hot-pressed composites exhibit an EMI SE difference of about +5 dB compared with their non-hot-pressed equivalents.

Ravindren et al. [81] investigated the effect of different loading levels of CB in a polymer blend (ethylene methyl acrylate EMA/EOC) on the EMI SE properties. They observed a preferential distribution of filler in the EMA phase; this behavior promotes a double percolation phenomenon that enhances electrical conductivity (Figure 7a). It also allows the lowering of the electrical threshold concentration compared to pure EMA. They concluded that a preferential distribution of fillers leads to superior electrical and shielding properties. The best shielding result, shown in Figure 7b, belongs to the composite with 30 wt.% loading (31.4 dB).

Mondal et al. [82] developed a strategy to produce chlorinated polyethylene/CB composites with a low percolation threshold. They used a commercial CB characterized by high conductivity. They found very promising results for 30 wt.% loading, characterized by an EMI SE of 38.4 dB in the X-band region and a thickness of 1 mm. By considering classical percolation theory, the researchers highlight that the electrical conductivity of the composite is quasi-two-dimensional.

#### 3.1.4. Biochar

Biochar is a carbonaceous material that derives from the thermo-chemical treatment of biomasses. Due to its properties, such as good electrical conductivity, thermal stability and high surface area, it is a valid candidate to substitute other carbonaceous fillers such as carbon nanotubes or carbon black. Moreover, its properties can be tuned by controlling process parameters such as temperature. Its electrical properties make it a good alternative to more expensive fillers for EMI shielding applications [105].

Li et al. [83] developed a high-performing composite made of UMWPE and biochar for EMI shielding purposes. They developed a process to produce composites with high loadings (up to 80 wt.%). By carbonizing the biomasses at 1100 °C, they obtained biochar with notable properties, such as high electrical conductivity (107.6 S/m). They performed electromagnetic measurements in the low frequency region of microwaves (from 300 MHz up to 1.5 GHz) and focused on the shielding effectiveness of the composites. The best result was obtained by the material with the highest loading level (80 wt.%): it exhibited an EMI SE above 35 dB across the entire frequency range.

Torsello et al. [84] focused on the impact of biochar on the properties of three construction materials. They dispersed inclusions of biochar (25 wt.%) in epoxy resin, polyethylene and cement and evaluated the electromagnetic properties from 100 MHz to 8 GHz. By applying the Looyenga rule, they extrapolated the permittivity of biochar and of matrixes; subsequently, they applied a proper model to calculate the shielding efficiency of composites with different loading levels in order to obtain the desired value of EMI shielding. Good results were exhibited by thick shields (30 mm) loaded with 30 wt.% of biochar.

Vijimon Moni et al. [85] focused on biochar and the influence of its activation on EMI shielding effect when dispersed in a PVA matrix. The activation process consists of a two-step chemical process aimed to partially modify the chemical structure of the biochar. They produced composite films with three different biochar concentrations: 1 wt.%, 3 wt.%, and 5 wt.%. Electromagnetic measurements were performed from 8 GHz to 20 GHz. Notable results were exhibited by the composite loaded with 5 wt.% of activated biochar: compared to the non-activated biochar, this material exhibits enhancements in EMI SE by 52% at 16 GHz and 40% at 20 GHz. At this frequency, the material shows an EMI SE above 40 dB.

Tolvanen et al. [86] demonstrated a way to produce environmentally friendly EMI shielding composites on a large scale through an easy process. They synthesized many materials with different compositions, consisting of a PLA matrix and graphite and biochar as dispersed phases. A large range of loading levels exhibited good shielding performances from 18 GHz to 26.5 GHz. Furthermore, excellent results were demonstrated by very thin films (0.25 mm) that showed an EMI SE above 30 dB.

### 3.2. Conductive Polymers

Composites are not the only possible solution for EMI shielding based on organic species, and conductive polymers represent an interesting solution [106]. These are organic materials characterized by a chemical structure able to conduct electrons. Conductivity can be tuned by controlling the synthesis parameters. As a result of this feature, their low cost, and good chemical resistance, they have been intensively studied as fillers for EMI shielding materials. In particular, researchers have mainly focused on two conductive polymers: polyaniline (PANI) and polypyrrole (PPy).

Lakshmi et al. [87] investigated the EMI shielding properties of PU/PANI at the S-band and L-band. By measuring the scattering parameters, they evaluated EMI SE: in the L-band, the composite exhibited poor shielding properties, whereas better properties were measured in the X-band (EMI SE > 10 dB, peak of around 26 dB at 9 GHz with a thickness of 1.9 mm).

A study by Oyharçabal et al. [88] focused on the impact of the aspect ratio of PANI fillers on the absorption properties of epoxy/PANI composites. Three different morphologies of PANI (globular, fibers, flakes) nanostructures were synthesized and mixed into epoxy resin in different compositions (5, 7.5, 10, 12.5, 15 wt.%). Electromagnetic measurements were performed in 2.4 ÷ 8.8 GHz and demonstrated the influence of the aspect ratio on the EMI shielding performances: flake-like PANIs exhibited the highest level of microwave absorption (peak of reflection loss RL of 37 dB), much greater than that of other morphologies (peak of RL of 8 dB).

In previous work, Tian et al. [89] focused on microwave-absorbing materials. They developed core–shell fillers based on PANI and PPy (PPy@PANI). PPy was used as the core micromaterial, whereas PANI acted as a nanoshell. They synthesized different shell thicknesses (from 60 nm to 120 nm). Electromagnetic analyses were performed on paraffin wax/50 wt.% PPy@PANI composites from 2 GHz to 18 GHz. The results showed the strong influence of shield thickness on the frequency of the absorption peak.

Similarly, Moučka et al. [90] investigated the influence of the PPy filler’s morphology on the EMI shielding of composites. Globules, nanotubes and microbarrels of PPy were dispersed in a silicone matrix in different compositions (1, 3 and 5 wt.%). The electromagnetic properties were evaluated in 5.85 ÷ 8.2 GHz (C-band). The composites exhibited a strong dependency of SE on electrical conductivity and aspect ratio, whereas the primary shielding mechanism was influenced by the morphology.

Gahlout et Choudhary [91] explored the use of PPy as a coating for MWCNTs in order to develop a polymer composite with broadband shielding properties. Through in situ polymerization of pyrrole in the presence of MWCNT, they were able to realize core (MWCNTs)–shell (PPy) structures. Different loadings of fillers were added to the thermoplastic polyurethane (TPU) matrix (10, 20, 30, 40 wt.%). Mechanical and electromagnetic measurements (in X-band) were performed. Composites with 10 wt.% loading exhibited good mechanical properties (improvement in tensile modulus and strength, similar strain at break compared with pure TPU) but poor shielding properties (EMI SE < 10 dB), whereas the best EMI SE was given by TPU/MWCNTs@PPy 40 wt.%: it exhibited an EMI SE greater than 40 dB. However, the mechanical properties have been significantly affected by the high level of loading (transition from ductile to brittle behavior).

### 3.3. Inorganic Fillers

The integration of inorganic fillers into polymer matrices has gained significant attention for enhancing the electromagnetic interference (EMI) shielding performance.

While metal-based materials are commonly used for reflection-based shielding, polymers are preferred for their ability to absorb electromagnetic radiation [9]. The effectiveness of polymeric nanocomposites for EMI shielding is influenced by various factors, including the surface area, porosity, aggregate structure, processing conditions, distribution of particles within the matrix, and filler loading level [107]. While metals have traditionally been used for EMI shielding, they are not without their limitations, including susceptibility to corrosion, wear, weight, and physical rigidity [28]. The integration of fillers in the polymeric matrix allows for the utilization of the unique physical properties exhibited by inorganic nanostructured materials, such as superparamagnetism and ferromagnetism, while mitigating the high manufacturing costs, challenging shaping and processing requirements associated with pure nanostructured materials [7].

MXenes, iron nanostructures, and silicon carbides have been extensively analyzed as inorganic fillers for EMI shielding. MXenes offer excellent electrical conductivity and mechanical properties, while iron nanostructures exhibit high magnetic permeability, enabling an effective absorption of electromagnetic waves [108]. Silicon carbides provide thermal stability and chemical resistance, making them suitable for high-temperature applications [109].

**Table 2 micromachines-15-00187-t002:** Inorganic composites for EMI shielding and their characteristics.

Filler	Matrix	wt.% Filler	Frequency (GHz)	EMI SE	RL	Thickness	Ref.
MExene foam	-	-	8 ÷ 14	70 dB	-	60 µm	[110]
Ti_3_C_2_T_x_	Epoxy resin	15	8 ÷ 12	41 dB	-	2 mm	[111]
Ti_3_C_2_ nanosheets	Paraffin wax	4	2 ÷ 18	-	40 dB at 8 GHz	-	[112]
Carbonyl iron fibers	Polyurethane foam	-	8 ÷ 18	-	25 dB at 10 GHz	3.5 mm	[113]
Iron NWs	Epoxy resin	50	0.1 ÷ 10	-	32 dB at 1.1 GHz	3.5 mm	[114]
Fe_3_O_4_ nanoparticles	Epoxy resin	20	4.5 ÷ 12	-	28 dB at 8.2 GHz	3 mm	[115]
Iron NWs	Paraffin wax	20	2 ÷ 4	-	29.7 dB at 5 GHz	3 mm	[116]
Fe_3_O_4_ nanorods	Paraffin wax	50	2 ÷ 4	-	40 dB at 2 GHz	8 mm	[117]
Flake-shaped carbonyl iron	Epoxy resin	50	8 ÷ 16	-	32.3 dB at 11 GHz	2 mm	[118]
NWs of bamboo-shaped SiC	Epoxy resin	-	2 ÷ 4	-	44.3 dB at 3.85 GHz	3.5 mm	[119]
SiC NWs	acrylic resin	3	8 ÷ 12	-	34.1 dB at 8 GHz	4.5 mm	[120]
SiC NWs 100 μm	Paraffin wax	20	8 ÷ 12	-	21.4 dB at 10.35 GHz	2.5 mm	[121]
SiC fibers@FeNi/C	Paraffin wax	15	2 ÷ 18	-	26.18 dB at 3.44 GHz	5 mm	[122]
BaTiO_3_ NWs	Epoxy resin	20	8 ÷ 12	-	24.5 dB at 9 GHz	3 mm	[123]
copper NWs	polystyrene	2.1	8 ÷ 12	35 dB	-	210 µm	[124]
SiO_2_@SiC	Paraffin wax	10	8 ÷ 12	-	54.68 dB at 8.99 GHz	4.49 mm	[125]

#### 3.3.1. MXenes

MXenes are a new class of two-dimensional (2D) graphene-like materials consisting of carbides, nitrides or carbonitrides of transition metals with a unique combination of properties. The general formula of MXenes is M_n+1_X_n_T_x_, in which M represents a transition metal element, X is carbon or nitrogen, n = 1–3, and T_x_ stands for a surface-terminating functional group, such as –OH, –F, –Cl or –O. The most investigated MXenes in this field of application are based on titanium carbides [112,126]. They were first discovered in 2011 by a team of researchers at Drexel University in Philadelphia, PA [127], and have since gained significant attention from the scientific community due to their potential for various applications, including in the field of electromagnetic interference (EMI) shielding [128]. MXenes have shown great potential as EMI shielding materials due to their high electrical conductivity and low electrical resistivity, making them effective at absorbing and reflecting electromagnetic waves [129]. In addition, they have a unique layered structure (Figure 8), which makes them highly flexible and conformable. This makes MXenes suitable for use as shielding materials in flexible and conformable electronic devices, such as wearable electronics and flexible displays.

Furthermore, MXenes have a high specific surface area [131] and can be functionalized with different groups, such as polymers or other organic molecules. This enables the tuning of their properties for specific applications and enhances their performance as EMI shielding materials. For example, functionalizing MXenes with polymers can improve their mechanical stability and flexibility, while functionalizing with other organic molecules can improve their thermal stability and chemical resistance. These properties make it a promising material for various applications. MXene has been used for energy storage, such as in supercapacitors, batteries, and fuel cells. Additionally, it has been used in electromagnetic interference shielding, water purification, and catalysis [132].

MXene is synthesized from the parent MAX phase, which is a class of ternary carbides and nitrides with a layered structure. Over seventy kinds of MAX have been discovered, and nearly twenty MXenes have been successfully synthesized [133].

MXenes offer distinct advantages over other 2D materials; in fact, their synthesis through selective etching of ceramic precursors allows for scalable production, making it suitable for industrial requirements [134]. The most common method used to prepare MXene is chemical etching of the MAX phase, with HF acid etching and fluoride salt etching being the most commonly used methods. In the HF acid etching method, the Al layer in the Ti_3_AlC_2_ MAX phase is selectively etched using HF acid, leaving behind the Ti_3_C_2_ MXene. The resulting MXene is terminated with functional groups such as –F, –O, and –OH because the etchant solution is rich in fluorine anions and hydroxyl groups [135].

In addition to these methods, the molten salt etching method has been proposed as a safer alternative to chemical etching. This method involves using a molten fluoride salt (LiF) to etch the MAX phase, which avoids the MXene being broken in hazardous environments. Li^+^ ions act as intercalation agents that enlarge the MXene layer spacing and weaken the interlayer interaction. This makes the multi-layer MXene more easily delaminated during sonication, resulting in single- or few-layer MXene flakes. Furthermore, a redox-controlled A-site etching of MAX phases in Lewis acidic melts has been proposed to synthesize unconventional MXenes with A elements such as Si, Zn, and Ga [33].

Sun et al. [136] conducted research on the fluorine-free partial conversion of Ti_2_AlC into carbide-derived carbon and Ti_2_CTx MXene through electrochemical etching in HCl. Further studies obtained fluorine-free electrochemical etching of aluminum from Ti_3_AlC_2_ using an aqueous electrolyte containing ammonium chloride and tetramethylammonium [137].

It is worth noting that the quality of the MXene synthesized is affected by several factors, such as the HF concentration, reaction time, reaction temperature, and MAX size. The surface functional groups of the resulting MXene also have a significant impact on its properties [138]. A study conducted by Mathis et al. [139] demonstrated the influence of MAX phase chemistry on MXene properties. A remarkable improvement in both the stability and the conductivity of MXene has been observed by incorporating an excess amount of aluminum during the synthesis of Ti_3_AlC_2_. This approach resulted in the formation of highly stoichiometric MAX and MXene phases. Simple adjustments to the etching conditions, such as altering the salt-to-acid ratio or introducing nitrogen gas during etching, can significantly impact the properties of the resulting MXene [140,141].

MXenes are also attracting attention as a potential alternative to graphene for providing an outstanding EMI shielding performance due to their exceptional metallic electrical conductivity. Nevertheless, the hydrophilic nature of MXene films can compromise their stability and reliability when used in moist or wet environments. Ji Liu et al. [110] presented an efficient and simple method for fabricating freestanding, flexible, and hydrophobic MXene foam with sufficient strength by assembling MXene sheets into films and then undergoing a hydrazine-induced foaming process. Unlike hydrophilic MXene materials, the resulting MXene foam exhibits hydrophobic surfaces and remarkable water resistance and durability. Additionally, the lightweight MXene foam shows an impressive improvement in EMI shielding effectiveness, reaching approximately 70 dB, compared to its unfoamed film counterpart (53 dB), due to its highly efficient wave attenuation in the favorable porous structure. As a result, the hydrophobic, flexible, and lightweight MXene foam with its exceptional EMI shielding performance holds great promise for applications in the aerospace and portable and wearable electronics industries.

Wang and co-workers [111] investigated the fabrication of epoxy-resin-based composites filled with Ti_3_C_2_T_x_. To further investigate the tunability of the filler’s electromagnetic properties, composites were made with the same filler annealed at 200 °C in a slightly reducing atmosphere (Ar + 5% H_2_). Electromagnetic measurements were carried out in the X-band. The annealing process was shown to be effective in partially removing surface polar groups, which caused the increase in conductivity and microwave shielding efficiency. The composites containing 15% wt. of annealed fillers achieved values of 105 S/m (σ) and 41 dB (SE), 176% and 37% higher than untreated fillers.

Luo et al. [112] used Ti_3_C_2_ nanosheets obtained by exfoliating Ti_3_C_2_T_x_ in the frequency range of 4 ÷ 18 GHz. For electromagnetic analysis, samples from 4 wt.% to 20 wt.% were prepared with paraffinic matrix. For the specimens loaded at 8 wt.%, in the whole K_u_-band (12 GHz ÷ 18 GHz), RL is less than 13.6 dB (very broad effective absorption bandwidth, or EAB). The effective absorption bandwidth (frequency bandwidth characterized by RL < −10 dB) increases further within frequencies below 10 GHz when the filler content is 12 wt.%. The authors suggest the possibility of intercalating magnetic fillers between Ti_3_C_2_ sheets in order to increase absorption properties at lower frequencies.

#### 3.3.2. Iron Nanostructures

The effectiveness of iron materials for microwave absorption has long been known, as shown in the examples of Radar Absorbing Materials (RAMs) used in the past [142]. Ferromagnetic metal particles that are optimized in shape are an intriguing subject of study because they possess higher saturation magnetization compared to ferrites and can attain a significantly large magnetic permeability in the gigahertz range [143]. When these nanoparticles are dispersed in a polymer-host matrix, they can demonstrate high microwave permeability and excellent absorbing performance. Moreover, magnetic nanoparticles with shape anisotropy, such as carbon nanotubes loaded with Fe nanowires, ferromagnetic alloy nanowires, and Fe, Ni, ZnO, and SiC nanowires, have been the focus of much research due to their high surface-to-volume ratio and aspect ratio [28,144].

These anisotropic magnetic particles may have a higher resonance frequency above Snoek’s limit in the gigahertz frequency range, which is due to their low eddy current loss stemming from their particle shape effects. In the study of Yu et al. [113], iron nanoparticles and nanowires are synthesized through the reduction of iron salts, specifically FeCl_3_·6H_2_O, with or without a parallel magnetic field. The results of this study provide a deeper understanding of the relationship between the shape of magnetic particles and their electromagnetic and microwave absorption properties, and shed light on the potential applications of these materials in various fields.

Therefore, research in this field has also been directed toward the development and study of iron-based nanomaterials. In a work by Li et al. [114], samples of epoxy resin loaded (at 50 vol.%) with two different iron-based structures (nanospheres and nanowires) were prepared. Measurements were conducted between 0.1 GHz and 10 GHz and showed that nanowires could be efficient for microwave absorption in the L-band (RL < −10 dB for t = 2 ÷ 4 mm). Given the large concentration of fillers, the absorption properties for lower levels of additives should be further investigated.

The adsorption properties of spherical Fe_3_O_4_ nanoparticles immersed in epoxy matrix were exploited by Zheng et al. [115]. Particle scattering was satisfactory, as was microwave absorption: in the frequency range 4.5 GHz ÷ 12 GHz for the sample containing 20 vol.% Fe_3_O_4_, an RL < −10 dB was reported.

Recently, Yang et al. [116] synthesized and characterized iron nanowires characterized by high aspect ratios with excellent absorption properties at frequencies included in the S-band. The tested samples (paraffinic matrix) had 15 wt.% and 20 wt.% filler. The minimum calculated RL value was −44.6 dB at 2.7 GHz (S-band). The best compromise between RL and thickness was achieved by the samples loaded at 20 wt.% (RL = −29.7 dB and t = 3 mm at about 5 GHz).

The study of Jazirehpour et al. [117] involved the preparation and analysis of magnetite nanospheres and nanorods using EDTA-assisted hydrothermal synthesis. Iron nitrate Fe(NO_3_)_3_·7H_2_O and EDTA were dissolved in water with a molar ratio of 1:9 (EDTA:FeNit), and a 2 M NaOH solution was gradually added to the solution until the desired pH was achieved. The pH was set at 13 and 8, respectively, to obtain nanorods and nanospheres. The obtained magnetite nanorods and nanospheres were mixed with paraffin (40, 50, and 60 wt.%) to prepare standard measurement toroid samples for electromagnetic measurements in a frequency range of 1–18 GHz at room temperature.

Authors reported that the aspect ratio of the nanorods and nanospheres played an important role in their microwave dielectric properties and static magnetic and microwave permeability. The real part of the nanorods’ permittivity was observed to be four times higher than that of the spherical nanoparticles. The coercivity of the nanorods was enhanced due to the higher surface and shape anisotropy. A shift in ferromagnetic resonance was observed in their microwave permeability. A significant percolative permittivity behavior was observed in the nanorods, especially in their real part of permittivity, while no such behavior was observed in the spherical nanoparticles. The nanorods and spherical nanoparticles produced a reflection loss peak value of about −40 dB at 2 GHz with thicknesses of 4 and 8 mm, respectively. Overall, the study concluded that with an increase in the aspect ratio of magnetite nanoparticles, their microwave attenuation coefficient increased, while their impedance match decreased. However, microwave absorbers with excellent absorption properties could be produced from high-aspect-ratio nanorods if their impedance matching could be improved.

Aspect ratio is an important parameter, which has been addressed by several research studies. On this matter, Yang et al. [118] prepared flake-shaped carbonyl iron (FCI) powder by ball milling spherical carbonyl iron (SCI) powder for 20 h. The milling process increased the aspect ratio from approximately 1 for spherical powder to an aspect ratio of 20–100 for flake-shaped powder (Figure 9). The thicknesses of the FCI particles ranged from 0.1 to 0.3 µm with a diameter of 5–10 µm.

The results showed that both the permittivity and permeability of the FCI powder were substantially increased after the milling treatment compared to the SCI powder. Specifically, the FCI particles exhibited higher permeability, higher permittivity, and better absorbing properties within the inspected frequency range. The magnetic loss tangent of FCI composites showed that FCI, with their large shape anisotropy, may have a higher resonance frequency and exceed the Snoek’s limit in the gigahertz frequency range. The FCI composites with 50 wt.% at 3 mm thickness had a calculated reflection loss of −23.0 dB at 5.5 GHz, while the corresponding SCI composites had a reflection loss of −11.6 dB at 10.4 GHz. The lower reflection loss of the FCI composites can be explained by the larger dielectric loss and magnetic loss than those of the SCI composites.

#### 3.3.3. Silicon Carbide (SiC)

Silicon carbide (SiC) is a widely used material for EMI shielding applications [145]. Based on different sizes, form factors and microstructures, one-dimensional SiC nanomaterials can be classified into nanowires (NWs), nanorods, nanowhiskers, nanobelts, nanotubes and nanocables [146].

Nanowires have a higher aspect ratio and larger specific area than nanorods and nanowhiskers. For this reason, among different microstructures, nanowires are the most promising one-dimensional SiC nanomaterials as EM microwave absorbers due to their high dielectric loss [147]. Several parameters (i.e., presence of defects, thicknesses, surfaces and interfaces) deeply affect the interface polarization and permittivity of SiC NWs [146,148]. As a result, it is necessary to control these factors by increasing the defect content, enlarging the surface or interface areas, forming a network structure, and selecting an appropriate thickness and appropriate temperature.

Relative to nanowires, the structure of greatest interest is the core–shell structure. Shells, for example, those of SiO_2_, BN and C, allow for improved surface polarization and conductivity of SiC, resulting in increased absorption properties and high chemical stability [149].

The morphology of SiC NWs (linear or bamboo-shaped) dominated by the vapor–liquid–solid growth mechanism can be controlled by the silicon vapor concentration, which is regulated by the vaporization temperature of the silicon source (Si and SiO_2_) in different sintering processes [119].

While the incorporation of uniform straight nanowires can enhance the mechanical properties of composites, the interfacial strength remains a challenge due to inadequate adhesion between the smooth nanowires and the matrix [150]. Straight NWs of SiC show a uniform and smooth surface with a low defect density. They can be prepared via CVD, carbothermal reduction, electrospinning and pyrolysis [151].

In contrast, bamboo-like nanowires offer distinct advantages for enhancing the properties of composite materials. The presence of concave and convex surfaces on these nanowires promotes stronger interfacial bonding with the matrix. This occurs through the formation of solid mechanical interlocking between the nodes of the nanowires and the surrounding matrix [152]. The uneven surfaces of bamboo-like nanowires also contribute to enhanced resistance against crack propagation [153]. SiC NWs with a bamboo structure have a rough surface with a high defect density. If straight SiC NWs are cleaned with HF or NaOH to remove the SiO_2_ shell, the SiC NWs will show a rough surface and a bamboo-like structure.

The planar defected bamboo-like structure shown in Figure 10 promoted the absorptive property of SiC NWs. This is due to the reduction in the permittivity and conductivity values together with the production of multiple scattering effects on the incident EM waves, thus increasing the absorption capacity of low-frequency waves [154]. As a confirmation of what was observed in the study by Zhang et al. [119], in which, with materials containing NWs of bamboo-shaped SiC, a minimum RL of 44.3 dB at 3.85 GHz with a bandwidth of 0.64 GHz (in the S-band) is achieved at a thickness of 3.5 mm.

Another interesting SiC NW fabrication technology is stereolithography, as reported by Xiao et al. [120], who fabricates an acrylic resin composite with SiC NW (3 wt.%). The minimum RL is −34.1 dB recorded at 8 GHz.

Kuang et al. [121] synthesized SiC nanowires of different lengths to study the effect of this feature on microwave absorption properties. The samples were made by embedding the filler in paraffin wax (20 wt.%). Nanowires with lengths greater than 100 µm exhibited the highest absorption capacity (RLmin = −21.4 dB compared with RLmin ≈ −10 dB of nanowires of length l = 1 ÷ 50 µm) and an absorption band of 3.5 GHz (in the X-band). Associated with a greater length of nanowires is the greater ease with which they make a conductive lattice (leading to greater energy dissipation related to conduction losses).

SiC is also used in a non-nanostructured form; in fact, in numerous cases, it is in fiber form. In a study by Guo et al. [122], it is possible to observe SiC fibers coated with a FeNi/C bilayer obtained via magnetron sputtering. In this case, good minimum reflection results (−26.18 dB) were obtained at low frequencies (3.44 GHz).

#### 3.3.4. Other Inorganic Structures

The absorption properties of barium titanate (BaTiO_3_) nanowires were evaluated by Yang et al. [123]. The fillers (Figure 11), synthesized via a one-step hydrothermal process, have an extremely high shape ratio (>1300, d ≈ 6 nm). To perform the measurements, the nanowires were loaded inside an epoxy resin. The maximum RL (−24.5 dB) was recorded at 9 GHz. The authors point out that given the great simplicity and productivity of the synthesis method, this methodology is also potentially industrially applicable for other 1-D ferroelectric nanostructures.

Al-Saleh et al. [124] developed a copper nanowire (CuNW)/PS composite powder with exceptional electromagnetic interference (EMI) shielding capabilities through solution processing. To create composites with a lower CuNW concentration, a dilution process was employed where the conductive powder was mixed with pure PS powder, followed by compression molding to form composite parts. The electrical percolation threshold of the composite produced through this process was only 0.24 vol.% CuNW, and electron micrographs showed that the conductive powder formed a segregated network within the polymer matrix. In the X-band frequency range, it was found that a 210 μm film made of PS composite containing 1.3 vol.% CuNW had an EMI shielding effectiveness (SE) of 27 dB, and PS with 2.1 vol.% CuNW had an EMI SE of 35 dB. Moreover, for 1.3 vol.% and 2.1 vol.% CuNW composites, approximately 54% of the overall EMI SE was contributed by absorption. The findings demonstrate that the CuNW/PS composite powder prepared via the dilution process and compression molding is a highly conductive material with excellent EMI shielding capabilities.

Xiang et al. [125] described the synthesis and characterization of silicon-based core–shell structures, the core of which is made with SiO_2_ particles and the shell with SiC (SiO_2_@SiC). Electromagnetic tests were conducted on paraffin samples (loaded at 10 wt.%) and showed good absorption properties. The minimum RL measured is −54.6 dB (at 9 GHz) and the maximum bandwidth is 8.5 GHz. It is worth noting that the samples that were subjected to reduction at a temperature of 1400 °C show discrete absorption (RL < −10 dB) even at frequencies belonging to the S-band.

### 3.4. Hybrid Fillers

In order to develop efficient, broadband and lightweight electromagnetic shields, numerous studies on novel hybrid fillers have been carried out [155]. The term “hybrid” refers to filler consisting of an organic (usually carbonaceous) and an inorganic component. By tuning the properties of these two components, one can modify shielding absorption and the reflection ability of the filler [42]. The organic part is useful to lighten the hybrid component and to act on electrical properties, whereas the inorganic one usually allows for more efficient tuning of the magnetic component. One of the primary classifications of hybrid materials for EMI shielding distinguishes between intercalation compounds and core–shell structures. Intercalation compounds refer to materials where guest molecules or nanoparticles are inserted between the layers of a host material, forming a layered structure. This arrangement provides enhanced EMI shielding properties due to the combined effects of the host material and the inserted components [156]. On the other hand, core–shell structures involve the formation of a core material surrounded by a shell material, which improves the EMI shielding performance by combining the unique properties of both the core and shell materials [157]. The core material typically possesses conductive or magnetic properties, while the shell material provides protection and stability. It is important to note that there are other types of hybrid structures for EMI shielding that do not fall into this specific classification.

**Table 3 micromachines-15-00187-t003:** Hybrid composites for EMI shielding.

Filler	Matrix	wt.% Filler	Frequency (GHz)	Max EMI SE (dB)	Thickness	Ref.
MXene/rGO	Epoxy resin	-	8.2 ÷ 12.4	50.7	3 mm	[158]
rGO/Mn_3_O_4_	Wax	25	2 ÷ 18	>20	5 mm	[159]
Fe_3_O_4_@PANI on Carbon fibers	Epoxy resin	-	8.2 ÷ 18	29	3 mm	[160]
Fe_3_O_4_ on CF@PANI	Wax	10	2 ÷ 18	-	2.7 mm	[161]
Fe_3_O_4_@C	PANI	10 ÷ 30	2 ÷ 8	65	1 mm	[162]
GO/MXene @ RC	-	-	X-band	>100	-	[163]
Co/Zn-MOF	Epoxy resin		X-band		200 µm	[164]
GNP/Fe_3_O_4_-C	-	-	X-band	28	0.12 mm	[165]
NF@Co/C	Wax	10	X-band	33	1.6 mm	[166]
rGO/MXene	Epoxy resin	1.2 (rGO) 1 ÷ 3.3 (MXene)	8.2 ÷ 12.4	55	0.5 mm	[167]
CNFs@ZrO_2_	Epoxy resin		K_u_-band	>90	0.72 mm	[168]
MWCNTs/Sendust	EVA/POE blend	Sendust: 100 phr MWCNTs: 5 ÷ 15 phr	0.1 ÷ 6	15	-	[169]
Graphene/NSF	PVDF	graphene: 3 NSF: 15, 30	1 ÷ 12	>50	250 µm	[170]
CuS/rGO	PANI	10 ÷ 50	300 KHz ÷ 3 GHz	>18	3 mm	[171]

#### 3.4.1. Intercalation Compounds

This family of hybrid fillers consists of layered structures. The peculiarity of these structures is the presence of intercalated particles (e.g., oxide nanoparticles) between the contiguous layers (e.g., MXenes, graphene and its derivatives).

A novel hybrid filler constituted by a 3D structure was synthesized by Fan et al. [158] They studied a strategy to realize a MXene-rGO structure by inserting MXenes nanosheets (Ti_3_C_2_T_x_) into GO. Further thermal treatment managed to convert GO in rGO and to create a hybrid foam. The researchers produced epoxy composites and characterized their electrical properties and electromagnetic behavior in X-band. The results demonstrated the shielding effectiveness of the MXenes/rGO foams in electromagnetic attenuation. They exhibited a maximum EMI SE of 50.7 dB with low sample thickness. In particular, the researchers highlighted the ultra-high specific SE exhibited by the sample with thickness of 1 mm.

Wang et al. [159] prepared a rGO/Mn_3_O_4_ filler via a one-step hydrothermal method for EMI shielding purposes. Mn_3_O_4_ nanoparticles were also synthesized to conduct a comparison. They mixed hybrid (rGO/Mn_3_O_4_) and inorganic filler to wax (mass ratio 3:1) to perform electromagnetic measurements in 2 ÷ 18 GHz. Composites with hybrid filler exhibited much higher EMI SE than inorganics: the calculated EMI SE was greater than 10 dB throughout the X-band, with a broad frequency band (about 6.5 ÷ 18 GHz) with EMI SE higher than 20 dB.

#### 3.4.2. Core–Shell

The term “core–shell” does not only refer to spherical structures, but also to other morphologies (e.g., fibrous structures, nanotubes, nanowires, etc.) that are made of two different materials. In general, core–shell nanospheres are characterized by an inorganic core and an organic shell, whereas high-aspect-ratio hybrid nanostructures are made of both of organic and inorganic core. Usually, the shell is useful to ensure better compatibility with the polymer matrix [14,172,173].

A complex hybrid filler for both electromagnetic shielding and structural purposes was synthesized by Movassagh-Alanagh et al. [160]. They deposited Fe_3_O_4_ nanoparticles on carbon fibers (CF) via electrophoretic deposition and subsequently coated Fe_3_O_4_ with a shell of PANI via the in situ polymerization of its monomer. Then, they added the filler to epoxy resin in different compositions. Electromagnetic measurements were carried out in 8.2 ÷ 18 GHz. The composite filled with 5 wt.% of CF@Fe_3_O_4_@PANI exhibited a maximum EMI SE of 29 dB with a thickness of 3 mm.

A system based on the same components but differently structured was synthesized by Zhang et al. [161]. First, they coated CF with a shell of PANI. Subsequently, they electrodeposited Fe_3_O_4_ nanoparticles on them. They focused on microwave absorption by measuring electromagnetic properties of wax composites (loading of 10 wt.%) in 2 ÷ 18 GHz. The sample with thickness of 2.7 mm exhibited a minimum reflection loss (RL) value of −46.86 dB, which corresponds to a very high level of absorption.

Manna et al. [162] proposed an easy method to synthesize Fe_3_O_4_@C core–shell nanostructures and to embed them in a conductive polymer matrix (PANI). The process consists of a hydrothermal step (formation of Fe_3_O_4_ nanoparticles) followed by a calcination treatment (formation of carbonaceous shell). Subsequently, hybrid fillers (10, 20, 30 wt.%, respectively to aniline) were added to a solution of HCl and aniline that polymerized via oxidant addition. Electromagnetic measurements in 2 ÷ 8 GHz highlighted the strong influence of the filler. Regarding the aniline ratio on shielding mechanism, the higher the ratio (30 wt.%), the greater the contribution of reflection (about 40 dB). Conversely, a low ratio (10 wt.%) guarantees a higher level of absorption (about 50 dB). Overall, the work highlighted very good shielding effectiveness for a 10 wt.% loaded composite (EMI SE ≈ 65 dB).

The study by Liu et al. [163] describes a new method for producing electrically conductive fibers made from transition metal carbide (MXene). Although MXene fibers are highly conductive, they are not very strong and are difficult to spin into usable forms. The authors of this study used a coaxial wet spinning process to produce MXene-based fibers with a tough outer layer made from regenerated cellulose (RC) and a conductive inner layer made from graphene oxide/MXene (GM). By optimizing the structure of the fibers, the authors were able to create a hollow RC@GM90 fiber with excellent mechanical and electrical properties. These fibers also demonstrated high performance in electromagnetic interference shielding and solar–thermal energy conversion. The authors also experimented with improving the spinnability of the MXene fibers by using a dispersion of graphene oxide (GO), which allowed them to produce solid GM@RC fibers with high conductivity and good electro-thermal energy conversion properties.

Li et al. [174] proposed a method to create an efficient hybrid absorber based on reduced graphene oxide, CNT and Fe_3_O_4_. They developed a three-step method in order to infiltrate an interconnected structure of rGO, CNT and Fe_3_O_4_ with polydimethylsiloxane (PDMS) and to fabricate thin films. The best performance was achieved by a composite made of four layers: it exhibited an RL of −50.5 dB at 13 GHz with a thickness of 1.42 mm. Furthermore, its EAB (corresponding to RL < −10 dB) was quite broad (5.7 GHz).

Wang et al. [175] synthesized hybrid nanospheres constituted by a magnetic core of Fe_3_O_4_ and a conductive shell of PANI. They performed the polymerization of an aniline monomer in the presence of Fe_3_O_4_ nanoparticles dispersed in a hydrochloric acid solution. They obtained Fe_3_O_4_@PANI core–shell nanospheres with different mass ratios (1:1, 1:2, 1:3) by changing the content of the aniline monomer. Consequentially, the study focused on the effect of the content of aniline monomer on the properties of materials. They separated the nanospheres and investigated their electromagnetic properties by preparing wax composites loaded at 50 wt.%. The researchers noticed a solid enhancement in absorption capabilities after coating Fe_3_O_4_ nanospheres with PANI. The material containing the hybrid filler characterized by a Fe_3_O_4_:PANI mass ratio of 1:3 showed a minimum RL of −55.5 dB at 17.3 GHz (corresponding to a thickness under 1.6 mm).

A core–shell structure made of a metallic core and an organic shell was synthesized by Wu et al. [176]. They proposed a multi-step method to produce a lightweight hybrid aerogel based on copper nanowires coated by graphene. Electromagnetic analyses were carried out from 8 GHZ to 18 GHz. The results demonstrated a SE superior to 40 dB over both the X-band and K_u_-band, with a predominant absorption mechanism. Moreover, this material exhibited good mechanical and electrical properties.

#### 3.4.3. Metal–Organic Frameworks (MOFs)

Metal–organic frameworks stand out as pioneering functional materials achieved through reticular chemistry, a strategy that allows for the assembly of molecular building blocks into precisely ordered structures connected by robust bonds [177]. Notably, MOFs exhibit extraordinary characteristics, boasting ultra-high porosity and expansive internal and external surface areas. What distinguishes MOFs even further is the extensive array of metal clusters and organic ligands, each with diverse geometries and connectivity [178]. MOFs are exceptional candidates across various applications, including catalysis, sensors, gas storage, batteries, water purification, and drug delivery [179]. Furthermore, their remarkable electrical conductivity and magnetic properties, coupled with favorable geometrical features facilitating multiple reflections due to high porosity and surface area, underscore MOFs’ outstanding performance in microwave absorption, marking them promising materials among electromagnetic absorbers [180,181].

A notable contribution comes from the work of Ma et al. [164], who successfully fabricated a carbonized wood composite film (CWF/EP/Co) with exceptional conductivity (105 S cm^−1^) and an outstanding EMI SE of 73 dB at a thickness of 200 µm. Through a meticulous process involving hot-pressing and the carbonization of wood soaked in a mixed solution of water-borne epoxy (EP) and Co/Zn-MOF, the CWF/EP/Co film demonstrated remarkable specific EMI shielding effectiveness in the X-band, surpassing all reported values for wood-derived EMI shielding materials. This innovative shielding film not only exhibits a high absorption but also demonstrates promising mechanical properties and effective Joule heating performance.

In another work, Fei et al. [165] employed a filtration-assisted self-assembly method, to produce composite films with a multilayer sandwich structure derived from a cellulose solution in ferric metal–organic frameworks (MOFs) (MIL-88B). The resulting C-MIL-88B/GNP films, incorporating graphene nanoplates (GNPs) and Fe_3_O_4_–C, exhibit satisfactory saturation magnetization and substantial conductivity. Particularly impressive is the EMI shielding effectiveness of 28 dB in the X-band frequency range for the composite film with five layers (0.12 mm). The superior microwave absorption and attenuation properties of the C-MIL-88B/GNP film arise from a synergistic effect involving dielectric loss, magnetic loss, and a unique multilayer structure.

Zhu et al. [166] have made a notable contribution by constructing a novel needlelike Co_3_O_4_/C array architecture from a metal–organic framework (MOF) precursor through a simple pyrolysis process. The resulting 3D hierarchical structure and multiphase components enable the architecture to provide high-efficiency electromagnetic interference (EMI) shielding, with the capability to shield up to 99.95% of electromagnetic radiation. Moreover, this architecture exhibits good thermal insulation and features fast ion transport channels, facilitating the construction of supercapacitors.

#### 3.4.4. Other Hybrid Materials

Similarly, Song et al. [167] developed a rGO/MXene hybrid filler and performed a more complex synthesis method. They tried to synthesize a shielding-effective filler characterized according to structural properties; they grew the hybrid filler on a Al_2_O_3_ honeycomb, used as a template. Then, they infiltrated it with epoxy resin and performed electromagnetic measurements in 8.2 ÷ 12.4 GHz. The composites loaded with the filler constituted by 1.2 wt.% of rGO and 3.3 wt.% of MXene showed the highest EMI SE (55 dB).

Deeraj et al. [168] focused on the detailed description of the process to create zirconia (ZrO_2_)-incorporated polyacrylonitrile (PAN)-based carbon-nanofiber-reinforced epoxy composites for use as an electromagnetic interference (EMI) shielding material (Figure 12). The process began with the electrospinning of zirconia-incorporated PAN nanofiber mats in N, a N-dimethylformamide solvent medium. These nanofiber mats were then thermally treated at 1000 °C to form zirconia-incorporated carbon nanofiber mats (CNFs), which were then impregnated with epoxy resin to create ZrO_2_/CNF epoxy laminas. The storage modulus of the ZrO_2_@CNF/epoxy composites was found to be higher than that of neat CNF/epoxy composites. The total EMI shielding effectiveness of the carbonized mats and CNF/epoxy composites was analyzed, and high specific shielding was observed. The stacking the ZrO_2_@CNF mats was shown to be a successful approach, and eight layers displayed an excellent EMI shielding effectiveness of over −90 dB in the K_u_-band.

Kim et al. [169] developed novel nanocomposites based on a polymer blend (EVA and polyolefin elastomer) and hybrid fillers. They focused on the effect of the addition of MWCNTs on the EMI shielding properties of polymer blend/Sendust composites from 100 MHz to 6 GHz. Sendust is a ternary magnetic alloy made of Fe, Si and Al. They observed that the addition of CNTs leads to additional losses due to different mechanisms (interfacial polarizations, dipolar polarizations). By increasing the content of MWCNTs, the EMI SE increases, as higher conductivity and permittivity cause enhanced dielectric loss.

Ahmed et al. [170] proposed hybrid fillers for EMI shielding applications based on graphene and nickel spinel ferrites (NSF). They studied the effect of NSF concentration on the shielding properties of polyvinylidene fluoride matrix, varying its content from 15 wt.% to 30 wt.% and fixing the graphene loading (3 wt.%), in 1 ÷ 12 GHz. They calculated EMI SE by considering the scattering parameters and found promising results for the composite loaded with 15wt.% of NSF. Higher concentrations of NSF lead to the agglomeration of the powder, causing a decrease in EMI SE. Absorption is the main shielding mechanism.

Hu et al. [171] prepared novel lightweight nanocomposites for EMI shielding based on a conductive polymer matrix (PANI) and hybrid filler (CuS/rGO). Firstly, they prepared the hybrid filler. Then, they performed an in situ polymerization of PANI with the presence of the filler (in different concentrations) that led to a uniform coating of CuS/rGO. The shielding effectiveness was calculated in 300 KHz–3 GHz; promising results were shown by the 20 wt.%-loaded composite, which exhibited an EMI SE higher than 18 dB in the whole frequency range.

## 4. Conclusions and Future Outlooks

The field of EMI shielding represents one of the most advanced frontiers in material science, requiring a shift towards rational materials’ design rather than a simple production–testing approach. The overwhelming necessity to produce new and better performing EMI shield materials is mainly related to the key sectors in which they are used, such as the aerospace and communication industries. These fields demand high EMI shielding performances together with low density due to strict weight limitations. Few materials can match the abovementioned requirements, and they are quite expensive and difficult to produce. These issues can only be neglected in performance-driven applications (i.e., satellites, aircrafts) but they must be faced when producing cheap and affordable devices (i.e., mobile phones, new generation vehicles). Accordingly, cutting-edge materials such as MXenes and graphene-related materials have been confined to a limited number of applications, while inexpensive inorganic materials and custom organic solution have attracted more interest year by year. Nonetheless, the EMI shielding material field is still far from groundbreaking and revolutionary advancements able to mark a solid step forward towards massive production of cheap and easily processable solutions. Nevertheless, we believe that the continuous research of innovative materials can lead the way toward a bright future for the material science of EMI shielding materials and related technological sectors.

## Figures and Tables

**Figure 1 micromachines-15-00187-f001:**
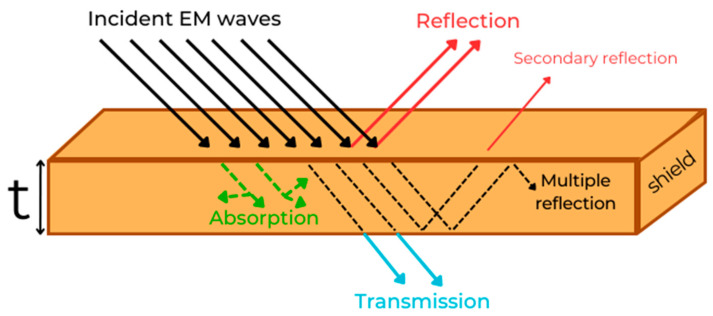
Schematic representation of microwave interaction with matter.

**Figure 2 micromachines-15-00187-f002:**
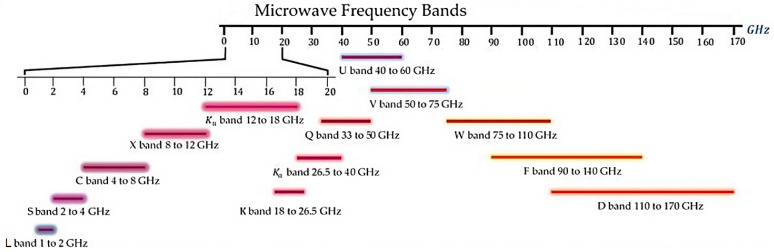
Microwave frequency region and sub-bands. Reprinted with permission from Ref. [36].

**Figure 3 micromachines-15-00187-f003:**
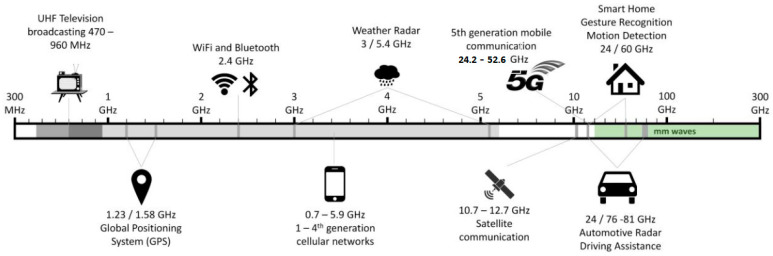
Applications of microwaves and their frequencies. Reprinted with permission from Ref. [50].

**Figure 4 micromachines-15-00187-f004:**
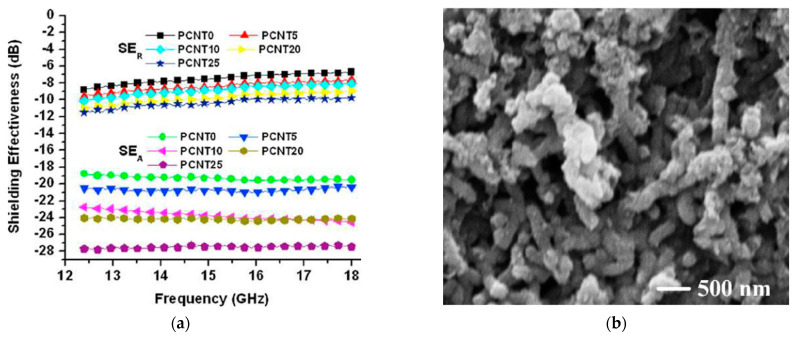
SE_A_ and SE_R_ contributions of MWCNTs/PANI composites (**a**) and micrograph of PCNT20, characterized by aniline to MWCNTs ratio of 4:1 (**b**). Reprinted with permission from Ref. [62].

**Figure 5 micromachines-15-00187-f005:**
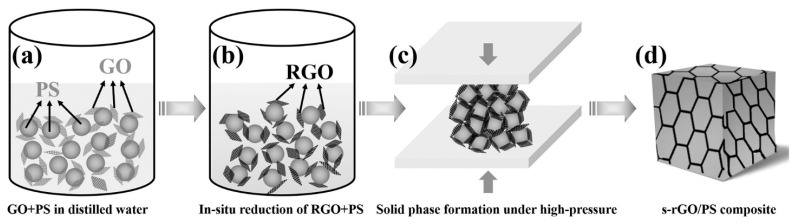
Scheme of production process of the composite. Suspension of GO and PS particles in distilled water (**a**), in situ reduction (with hydrazine hydrate) of GO (**b**), solid-phase compression molding of PS/rGO after water removal (**c**), and final composite material (**d**). Reprinted with permission from Ref. [75].

**Figure 6 micromachines-15-00187-f006:**
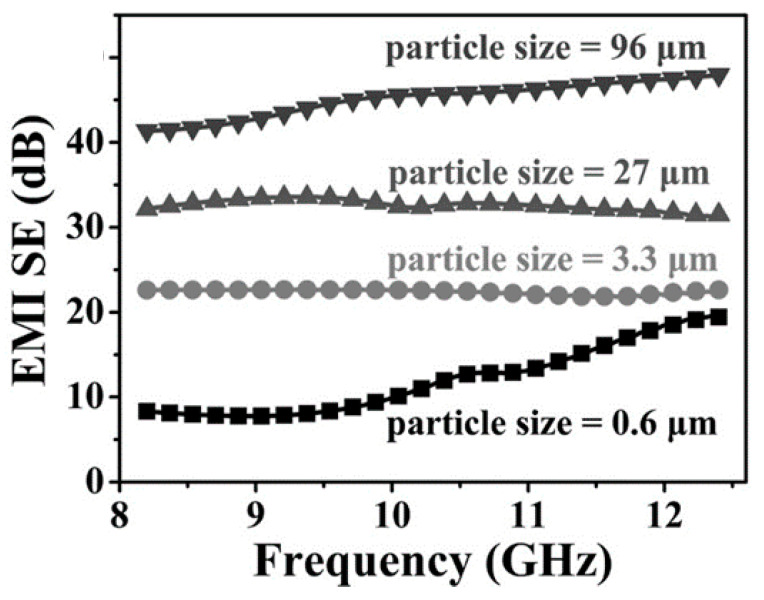
EMI SE of rGO/PS composites with various particle sizes. Reprinted with permission from Ref. [75].

**Figure 7 micromachines-15-00187-f007:**
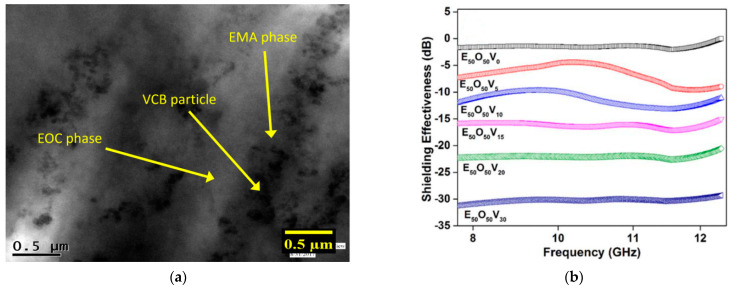
Preferential distribution of CB in EMA phase (**a**) and EMI SE of composites with different CB loadings (**b**). Reprinted with permission from Ref. [81].

**Figure 8 micromachines-15-00187-f008:**
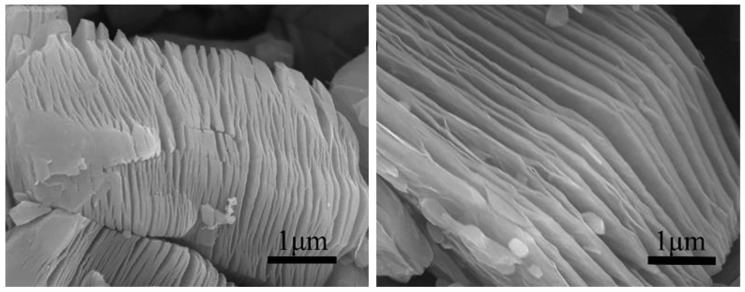
SEM image of MXenes’ layered structure. Reprinted with permission from Ref. [130]. Copyright 2016 American Chemical Society.

**Figure 9 micromachines-15-00187-f009:**
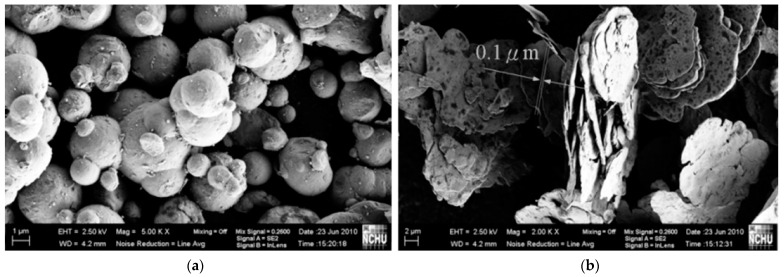
Spherical carbonyl iron (**a**) and flake-shaped carbonyl iron (**b**). Reprinted with permission from Ref. [118].

**Figure 10 micromachines-15-00187-f010:**
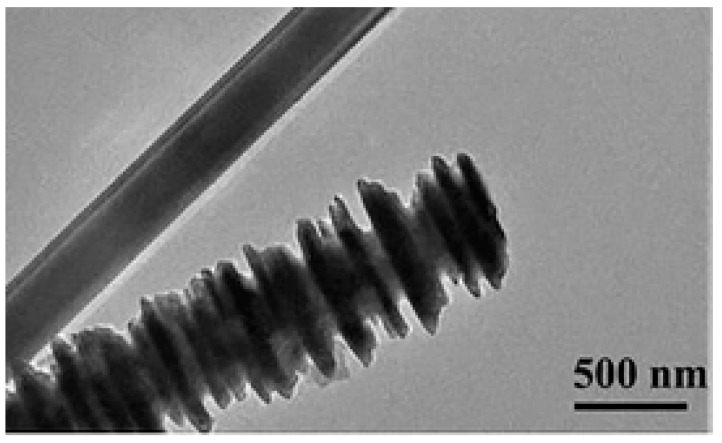
FESEM image of linear and bamboo-shaped SiC NWs. Reprinted with permission from Ref. [119].

**Figure 11 micromachines-15-00187-f011:**
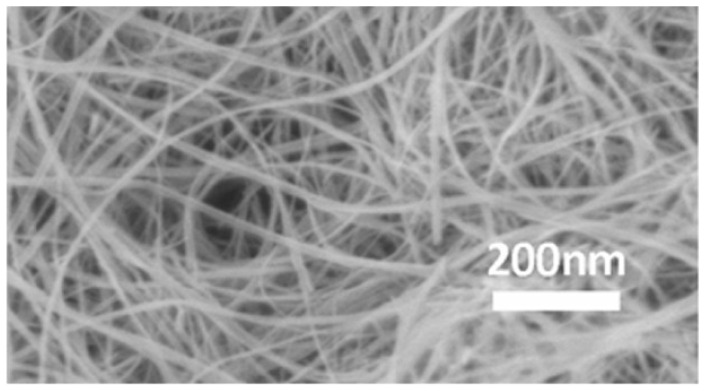
Nanowires of BaTiO_3_. Reprinted with permission from Ref. [123]. Copyright 2013 American Chemical Society.

**Figure 12 micromachines-15-00187-f012:**
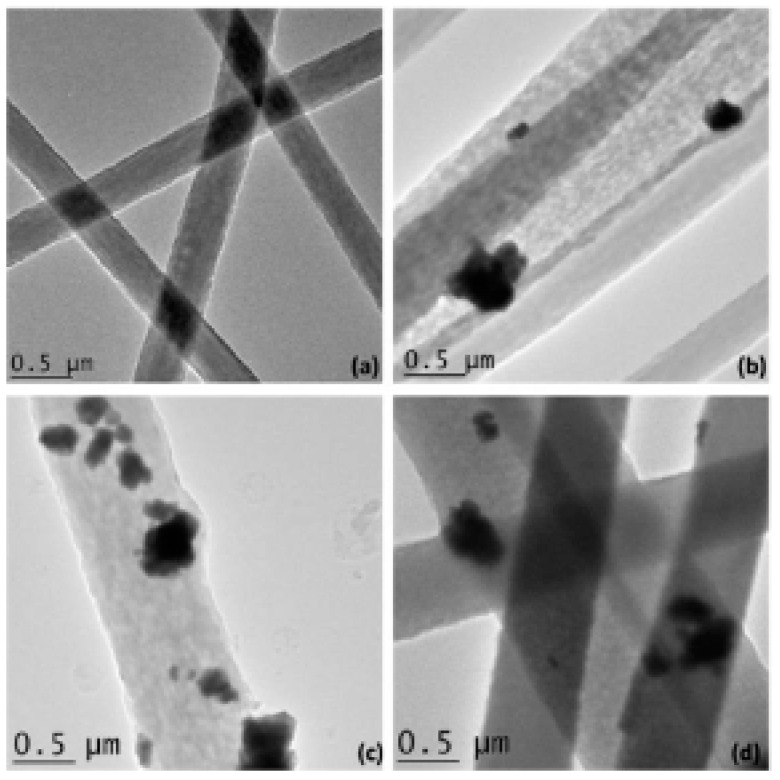
TEM micrographs of PAN/ZrO_2_ electrospun fibers coated on copper grid. (**a**) PAN alone, (**b**–**d**) PAN/ZrO_2_. Reprinted with permission from Ref. [168].

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
