# Peer review of "A Comprehensive Review of Electromagnetic Interference Shielding Composite Materials"

_micromachines, 2024, doi:10.3390/mi15020187_

Round 1
Reviewer 1 Report
Comments and Suggestions for Authors
This manuscript summarizes the latest research advances in composite materials used in the field of electromagnetic shielding. I would recommend this work to be accepted with the following issues to be addressed.
1. The microscopic mechanisms of electromagnetic shielding should be discussed in more detail. For example, what factors affect the macroscopic properties? What is the constitutive relationship between microscopic morphology and macroscopic performance?
2. How do the different structures from 0 to 3 dimensions affect the performance?
3. In the core-shell section, more supporting information from the literature should be provided.
Comments on the Quality of English LanguageThe English in this manuscript should be approppriately corrected.
Author Response
Please see the attachment.
Translator
Translator

Reviewer 2 Report
Comments and Suggestions for Authors
The manuscript proposes a good overview of materials for EMI shielding.
However some aspects are missing or should be clarified in a revised version. My recommendations are listed below:
- In many places the role of magnetic materials is cited to be involved in EMI. However, the mechanism relating magnetic properties to EMI is not explained clearly, though it is well known that magnetic materials help decrease the wave impedance, reducing the reflection and favoring absorption. This fact should be clearly explained.
- Regarding absorption, it is not sufficiently cited in the review. Indeed absorption is of great importance for EMI shielding since it is preferred to reflection that simply reflects the signal in the surroundings, inducing secondary spurious interference detrimental to electronic devices, etc… Absorption truly cancels the signals. I recommend that the advantages of absorption for EMI are better put in evidence in the revised manuscript, for example by adding a column in each table, specifying if EMI is mainly induced by reflection, or transmission, or both. If significant absorption is achieved, the thickness of the shield must also be indicated in each table.
- Related to that, a specific section should be devoted to absorption for EMI shielding, including namely the fact that porous materials and foamed composites help to facilitate EMI shielding due to absorption.
- The review should also include MOF structures (Metal-Organic-Framework) since such structures are now frequently reported for EMI induced by absorption
- Finally the review should also include a section discussing to what extend the various processes presented for the fabrication of materials for EMI are green.
- The sentence in the abstract : “The selective shielding of microwave represents 20 the only way to achieve the control on crucial technological sectors allowing the further step for-21 wards in a safe and secure society” must be clarified.
- The acronym EAB at line 673 must be defined
- Jf in equation 4 is never defined
- RL in table 2 is not defined in the main text
Comments on the Quality of English Languageno particular comment
Author Response
Translator
Please see the attachment.

Round 2
Reviewer 2 Report
Comments and Suggestions for Authors
Authors have addressed my comments properly.